# Cyber-Physical System for Smart Traffic Light Control

**DOI:** 10.3390/s23115028

**Published:** 2023-05-24

**Authors:** Siddhesh Deshpande, Sheng-Jen Hsieh

**Affiliations:** Engineering Technology and Industrial Distribution Department, Texas A&M University, College Station, TX 77843, USA; sid_2o@tamu.edu

**Keywords:** cyber-physical system, machine learning, smart traffic lights

## Abstract

In recent years, researchers have proposed smart traffic light control systems to improve traffic flow at intersections, but there is less focus on reducing vehicle and pedestrian delays simultaneously. This research proposes a cyber-physical system for smart traffic light control utilizing traffic detection cameras, machine learning algorithms, and a ladder logic program. The proposed method employs a dynamic traffic interval technique that categorizes traffic into low, medium, high, and very high volumes. It adjusts traffic light intervals based on real-time traffic data, including pedestrian and vehicle information. Machine learning algorithms, including convolutional neural network (CNN), artificial neural network (ANN), and support vector machine (SVM), are demonstrated to predict traffic conditions and traffic light timings. To validate the proposed method, the Simulation of Urban Mobility (SUMO) platform was used to simulate the real-world intersection working. The simulation result indicates the dynamic traffic interval technique is more efficient and showcases a 12% to 27% reduction in the waiting time of vehicles and a 9% to 23% reduction in the waiting time of pedestrians at an intersection when compared to the fixed time and semi-dynamic traffic light control methods.

## 1. Introduction

The *Manual on Uniform Traffic Control Devices* (MUTCD) [1] states that traffic lights are an important tool for improving road safety, reducing traffic congestion, and optimizing the use of roadways. Traffic lights are used to manage and control the flow of vehicular and pedestrian traffic at intersections. By alternating the signal between green, yellow, and red, traffic lights create a system of order and direction that reduces the likelihood of collisions and accidents. They also allow pedestrians to safely cross the road by stopping traffic and giving them the right of way.

While traffic lights are an effective means of enhancing the safety of people at intersections, they have the unintended consequence of increased travel times for drivers and pedestrians who must halt at red lights. Lv et al. [2] noticed that the likelihood of people halting and waiting at an intersection increases as the number of road crossings grows. The traffic lights are designed to manage conflicting traffic flows, meaning they often prioritize one direction of traffic over others, according to the *Signal Timing Manual* (STM) [3]. This strategy can delay vehicles or pedestrians waiting to cross the intersection. For example, if more cars are on one road than another, the traffic light may give a longer green light to the road with more traffic, causing delays for the other route. The *Highway Capacity Manual* (HCM) [4] states that the travel times for vehicles and pedestrians increase if there is a lack of coordination between different intersections. In some cases, the timing of traffic lights may not be synchronized, resulting in vehicle stop-and-go traffic and longer wait times for pedestrians.

The shortcomings of the simple traffic light control system led researchers to investigate smart traffic light control systems. The term “Smart Traffic Light Control System” refers to employing sensors and algorithms to control and optimize vehicular traffic. Real-time traffic data analysis allows the system to fine-tune traffic lights and improve traffic flow to minimize congestion and maximize safety. The control system may use machine learning models and optimization techniques to reduce travel time or to find the relation between various traffic signal parameters. An intelligent traffic light control system is generally more advanced and complex than traditional ones. Overall, an intelligent traffic light system aims to manage and control traffic signals as per the traffic volume at intersections. Intelligent traffic light control systems may also include anomaly detection, passage to emergency vehicles, connection to public transport systems, and integration with other smart city technologies.

## 2. Literature Review

The difficulties of urban traffic and intersections have prompted several proposed remedies from researchers in recent years. Ghazal et al. [5] proposed a system that uses infrared sensors and microcontrollers to control traffic lights at simple two-way intersections based on vehicle density. Diaz et al. [6] built an intelligent traffic light system with a Raspberry Pi, PIR sensor, and LED traffic light for a simple, single-lane, two-way intersection. These two approaches consider only single-lane traffic. Additionally, the traffic modeling is not specified for the performance evaluation part.

Silva et al. [7] proposed intelligent traffic lights for low-traffic conditions. This work presented a methodology that used a LanPro module installed on the vehicles to decide the path and avoid the red traffic signals. However, as mentioned in the paper, this method is only suitable for low traffic scenarios. Alharbi et al. [8] showcased a dynamic traffic light management system, which increases the green light time if more vehicles are detected but does not explicitly mention if the green time is reduced if fewer vehicles are detected.

Sen and Head [9] proposed a system that can skip green lights for the approaches/roads without vehicle flow during a traffic light cycle. This technique is better when the traffic is less on a particular route, and the traffic volumes do not change very often. Li et al. [10] proposed a methodology to optimize the fixed traffic light timings on isolated intersections. They optimized the traffic signal plan to reduce weighted vehicle and pedestrian delays. However, their system is not smart, which will detect pedestrians and vehicles in their respective lanes and adjust the traffic signal time as per the detected traffic.

Younis and Moayeri [11] worked on the simulation of dynamic traffic lights with one method based on sensors and another based on communication between vehicles. This work shows the results based on the through-vehicle movement but does not consider turning vehicles at the intersection. Tchuitcheu et al. [12] considered adjusting traffic lights based on the waiting queue of vehicles at the intersection in incoming and outgoing lanes. Gandhi et al. [13] also considered a smart traffic light system, which calculates the green traffic light timing based on vehicles detected by the camera sensor but does not consider the phases which allow the signaling of green light to non-conflicting movements from different incoming streets.

Pratama et al. [14] proposed a system to change the green light timings based on the vehicle density in incoming and outgoing lanes. This method is effective when there are a bunch of intersections close to each other, affecting the traffic density on the streets. Chavan et al. [15] presented a sensor network with embedded technology to manage traffic flows. In this work, they did not clearly mention the traffic light parameters for their technique’s performance evaluation.

Hirankitti et al. [16] showcased an agent-based intelligent traffic light control model. In this work, they used rules which will perform actions based on inputs such as current traffic phase, traffic light timing, the queue length of vehicles, and incoming and outgoing vehicles at an intersection and outline lane space availability, but did not mention the sensor type or how frequently and when the data will be collected. Almawgani [17] used a technique that applies different image processing algorithms for nighttime and daytime traffic detection. However, the work built a simple prototype to showcase the working of an image processing system and did not include modeling traffic flows at an intersection.

Li et al. [18] used cameras to capture images or videos of traffic conditions and applied machine learning algorithms to analyze and interpret the data. Wiering et al. [19] simulated an intelligent model using reinforcement learning to optimize the traffic light setting. They evaluated the performance of their technique on a group of intersections by making decisions based on vehicles at the traffic intersection. This method is more focused on optimizing traffic light parameters and, thus, there is no mention of the sensors to collect the data.

Wei et al. [20] also used deep reinforcement learning to optimize traffic light timing. Their system minimized the delays and improved traffic flow for through-movement vehicles but did not consider the left-turning vehicles. Liu et al. [21] presented a multi-agent Q-learning approach to control the traffic lights at the intersection. This method mentions adapting to pedestrians, but the algorithm does not employ the technique when pedestrians are detected at the intersection. Linag et al. [22] proposed a method that uses deep reinforcement learning with multiple optimization elements. Their system improved the vehicle waiting time by 20%. Their system inputs are vehicle position and vehicle speed but do not include when and where the data points are captured and what should be the frequency of data capturing in real-time. Göttlich et al. [23] utilized linear programming optimization to determine the optimal traffic light timing for a given set of traffic conditions. Park et al. [24] proposed genetic algorithms to optimize specific traffic signal parameters only for high-traffic cases at intersections.

Genders and Razavi [25] proposed a technique using reinforcement learning and a convolutional neural network to optimize the traffic light timings for an intersection. The inputs to the neural network include the presence of the vehicles in a specified area, vehicle speed, and the current phase of traffic lights. Oliveira et al. [26] used multiple neural networks to optimize the traffic light timings. This method gives the same inputs to different ANN, which predicts the same output. This work focuses more on neural network performance and does not specify parameters such as traffic flows or traffic light cycles for an intersection. Abbas et al. [27] presented a high-accuracy controller that can change the next phase timing and net phase green light time based on the current phase data collected from the roadside data collection (RSDC) units but did not mention the exact type of the sensors and the frequency of data collection. Additionally, the system evaluation does not mention the details of traffic light phases.

McKenney and White [28] applied an approach to control traffic signal lights based on the number of vehicles in a section of a road near the intersections. The approach considers various parameters related to data collection to decide the traffic light switching and green light timing change. Zhou et al. [29] used a wireless sensor network to detect the traffic at multiple intersections. This work used a technique that adjusted the traffic light phases and timing based on detected vehicles’ information. Piris et al. [30] also proposed a technique to optimally place the wireless sensors in a network of intersections. Additionally, this work presented a traffic light management system that focused on controlling multiple intersections and efficient communication of messages in a sensor network.

Navarro-Espinoza et al. [31] applied various machine learning models to predict the traffic flow for intelligent traffic lights. Muntean [32] proposed a multi-agent system to estimate the traffic volumes at junctions and car parking. Neither approach presented any control strategy for the traffic lights. Artega et al. [33] proposed a fuzzy logic method to control traffic lights based on the flow rate of vehicles. Nimac et al. [34] presented pedestrian detection and a traffic light control scheme using radars, but this technique is focused on the traffic light trigger mechanism and is not optimizing the pedestrian traffic lights.

Table 1 summarizes the details of the methodologies discussed in the literature review. From the table, it is clear that many researchers tend to give weightage to one factor: solving the congestion of vehicles and optimizing the traffic light systems using various techniques and tools. These methodologies generally focus more on the vehicle parameters that impact vehicle travel time. They try to reduce the delays at the intersection using sensors and algorithms. Moreover, many approaches involve modeling the traffic flow through an intersection and then finding the timing that minimizes some measure of congestion, such as travel time or delay. They apply machine learning and optimization techniques to reduce vehicle delay, but reducing pedestrian delay is not considered.

Thus, this paper’s objective is to propose an intelligent traffic light system and dynamic traffic interval technique that considers both vehicle and pedestrian traffic volumes to minimize waiting time at intersections. The next sections of this paper will describe the proposed system’s design, implementation, and evaluation.

## 3. Methodology

The traffic lights control the vehicle and pedestrian flow at intersections where traffic travels in different directions. The traffic flows moving in various directions are called movements and are categorized as left-turn, through, right-turn, and pedestrian movements, as shown in Figure 1. To differentiate the traffic flows, each movement is identified by a separate number (HCM) [4]. The traffic signal phases are then used to group certain movements, which allows the traffic to move in an orderly manner.

Figure 2 depicts a ring and barrier diagram for an 8-phase intersection. In this diagram, rings organize phases so that they are synchronized without interference and consist of a series of competing stages. At barriers, phases in both rings conclude concurrently. This enables dual or two-ring operations, allowing compatible phases to run concurrently with those of the opposite ring. The phases are an important part of the system as they control the timing and sequence of the green, yellow, and red lights for the incoming traffic movements at the intersection. Typically, they distinguish between major and minor street segments. Additionally, the movements are categorized as permitted and protected movements.

### 3.1. Dynamic Traffic Intervals

As the number of vehicles or pedestrians waiting at the intersection increases or decreases, the time required for them to pass through the intersection changes. Thus, the traffic light timings need to be appropriately planned so they do not cause delays at the intersection. To effectively manage the traffic flow at the intersection, this work proposes a dynamic traffic interval technique where the vehicle and pedestrian volumes waiting at a red light at the intersection are divided into low, medium, high, and very high categories. This is done to efficiently use the traffic light timings for all the phases and movements. Dividing the traffic into intervals allows setting the time of the green light only for the vehicles and pedestrians that request to cross the intersection. This work selects the four categories of traffic based on the simulation parameters and results. Each category represents a group of vehicles waiting at an intersection at a red light. The number of vehicles can be different for each category for different intersections and depends on the demand for the service requested by the user. To establish the categories of traffic, it is important to perform a critical movement analysis and define the detection zone at an intersection, as explained further in this section.

Figure 3 shows dynamic traffic light timing distribution for an 8-phase intersection. Each phase has green light timing for four categories of traffic. The timing of the green light can be adjusted as per the traffic category detected. Based on the green light timing of an ongoing phase, the red-light timing for other phases and the total cycle length will change. The cycle length is the time it takes for a traffic signal to complete a full cycle of all its signal phases. The yellow clearance time alerts drivers that the right-of-way assignment at the intersection is going to change. The red clearance time allows vehicles that entered the intersection during the yellow change period to reach a safe position before the next phase begins (STM) [3]. HCM [4] has suggested the red and yellow clearance interval timing for different vehicles approaching speeds. The timing for both lights is referred to as the change period. This change period is fixed and is based on the average approaching vehicle speed at the intersection.

Critical movement analysis is performed to calculate the traffic light timings. Critical movement analysis is a methodology used to identify the movements of vehicles that have the highest potential for conflict at an intersection. Critical movement analysis aims to evaluate phasing requirements and signal timing parameters. This analysis uses conflicting phases, often a left-turn phase and an opposing through-movement phase, to identify the crucial phase pairings. To determine the critical phase pair, the total volume of cars for all sets of conflicting phases is compared. Figure 4 shows the vehicle volumes associated with each phase. The following equations determine the critical volume for an intersection.
(1)$$CPP_{1}=\max\left(v_{1}+v_{2},\;v_{5}+v_{6}\right)$$
(2)$$CPP_{2}=\max\left(v_{3}+v_{4},\;v_{7}+v_{8}\right)$$
(3)$$CV=CPP_{1}+CPP_{2}$$
where $CPP_{1}$: Critical phase pair for street 1.$CPP_{2}$: Critical phase pair for street 2.$v_{1-8}$: Critical lane volume for phases 1 to 8 (vehicles/h).CV: Critical volume for an intersection.

**Figure 4 sensors-23-05028-f004:**
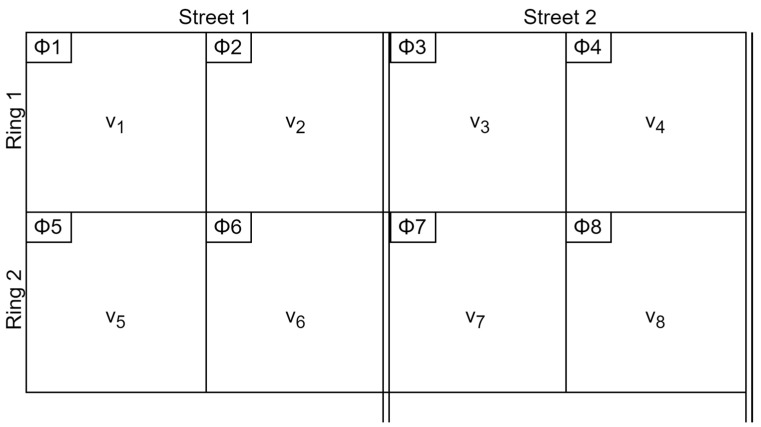
Vehicle volume for different phases.

This critical volume is then divided into four volumes where the traffic is categorized as low, medium, high, and very high traffic. Thus, using the critical volumes for each category, the four different cycle lengths are calculated by the following equation based on Webster’s Least delay cycle [35]. The cycle length will be fixed for each of the four categories as it is based on the critical volume for an intersection. This equation considers the saturation flow rate of 1900 passenger cars per hour per lane as stated by HCM [4].
(4)$$C=\frac{1.5L+5}{1.0-Y}$$
where C: Optimum, minimum delay cycle length (s).L: Lost time per cycle (s).Y: Sum of the critical lane volumes divided by saturation flow rate.

Then, the timing of the green light for each phase is estimated using Equations (5) and (6) [36] for four traffic conditions. This time is determined based on the expected cycle length and critical movement analysis.
(5)$$A_{t}=C-\left(\sum CP_{i}\right)$$
(6)$$G_{i}=\frac{V_{A}}{V_{T}}\times A_{t}$$
where $A_{t}$: Available time to apportion between all phases’ green interval (s).C: Calculated cycle length (s).$CP_{i}$: Change period (yellow change interval plus red clearance interval) for each phase (s).$G_{i}$: Phase green interval for each phase (s).$V_{A}$: Critical lane volume for phase *i* (vehicles/h).$V_{T}$: Sum of critical lane volumes for all phases (vehicles/h).

The green light timing for each traffic category is based on the sensor detection zone. The detection zone is the area where the sensor collects the data. For this research, the cameras are considered as sensors installed at the intersection. Cameras provide data-rich information, and recent advancements in machine learning models prove that vision systems are effective in detecting traffic. This work assumes that cameras will have a clear view of the lanes and will detect the traffic at night when the lights are installed at the intersection.

The occupancy zone is divided into four segments, as shown in Figure 5 and Figure 6 for vehicles and pedestrians, respectively, according to the critical volumes for each category of traffic. Each segment represents the number of vehicles/pedestrians that will be waiting at the red light with respect to the critical volume. The traffic will occupy a certain area of the zone and the number of vehicles and pedestrians will be calculated based on these four segments. For the pedestrians, the cameras will take the image of curbside where pedestrians wait on a red signal. The traffic category and its respective occupancy percentages are given in Table 2. The 100% occupancy is the area under the detection where the camera sensor can clearly see vehicles and pedestrians.

### 3.2. Framework

Figure 7 depicts an overview of the framework. The proposed framework is organized into three systems: physical, cyber, and control. The physical system consists of a junction where two or more streets cross, vehicles and pedestrians who wish to cross the junction, sensors that collect data, traffic lights that allow or prevent vehicles and pedestrians from crossing, and a programmable logic controller that controls the traffic lights. The cyber system employs machine learning (ML) algorithms to predict traffic conditions and light timings, reducing delays for pedestrians and automobiles. The control system, which includes a ladder logic program, influences the physical system by altering the on and off times of the traffic lights.

The operation of the smart traffic light control system is given in Figure 8. The proposed system’s role is to detect traffic conditions and regulate intersection traffic lights. The sensors are cameras that supply the information required by the cyber system. The CNN model determines the traffic category, while the ANN/SVM predicts green light timings based on traffic photos, vehicle count, and pedestrian count. After that, the anticipated values are saved in the DDE client file, which communicates with the DDE server and ladder logic. To eliminate delays, the smart traffic light control system can adjust the timing of the green and red lights based on predictions. MATLAB was used to create machine learning models. Visual Basic for Applications (VBA) was used to convey data to ladder logic, and the control system runs on the RSLinx Classic and RSLogix platforms.

#### 3.2.1. Physical System

This work focuses on an intersection to reduce vehicle and pedestrian travel delays. The traffic intersection is a central part of the physical system and framework. Traffic intersections can be regulated by traffic lights, stop signs, yield signs, or other traffic control devices to ensure safe and efficient traffic flow. They are critical components of transportation systems in urban areas and are designed to manage traffic movements and minimize the risk of collisions and accidents.

The collector roads are common in cities, and they connect local streets to arterial roads and are designed to provide access to local destinations, such as homes and businesses. This research considers a four-way junction on collector roads, where many users can approach the intersection at once. The junction has two streets crossing each other. Each street can have one or more incoming and outgoing vehicle lanes and two lanes for pedestrians on either side. The pedestrians can cross the junction using marked crosswalks at the intersection.

The traffic lights are usually installed at the intersection on poles or mast arms over the roadway. The lights face each lane, indicating when vehicles and pedestrians are allowed to move and when they must stop. The location and placement of traffic lights at intersections are carefully planned to convey the information and right-of-way to the users clearly. The MUTCD [1] recommends the traffic light types and installation procedures based on the intersection characteristics. This work considers the green, yellow, and red lights for each lane. Additionally, a flashing yellow light is considered for the permitted movements of the vehicle.

A traffic detection camera is the most common type of sensor utilized at intersections to detect traffic. The cameras are usually installed on the traffic poles with a certain height to get a specific view of the lanes. The camera specification is chosen based on the demand for traffic signal service and local jurisdiction guidelines TCM [4]. In this work, the cameras are programmed to capture images of the lanes when the traffic light changes from red to green. Then, the captured photos of the intersection are sent to computer vision algorithms to estimate traffic patterns and volume.

Traffic light controllers are electronic devices that control the operation of traffic lights at intersections. These are the cabinets that can be installed in a central control room, allowing operators to monitor and adjust signal timing across multiple intersections in real time. Alternatively, controllers can be distributed across individual intersections and communicate with each other to coordinate signal timing and optimize traffic flow across a wider area. The controllers are programmed to control the on/off times of all the traffic lights at the intersection. Sensors installed at the intersection continuously communicate real-time information to the controllers. The controller then sends the data to the cyber system for further processing.

#### 3.2.2. Cyber System

The cyber system consists of machine learning models to process the sensor data. In this work, the four traffic categories are predicted using different kinds of pre-trained CNN models. In CNN models, the images are represented as numeric matrices where a single image is divided into pixels, and each pixel is assigned a numeric value. These matrices are processed by CNN models using filters or kernels, which have been trained to identify features. This process is known as convolution. The convolutional layer applies numerous convolution filters to the image. Then, the pooling, or subsampling layer acts to decrease the spatial dimensions of the data, which helps to reduce computational needs and control overfitting. Finally, the fully connected layer, functioning like a traditional multi-layer perceptron, usually employs a SoftMax activation function to classify the input image into different categories based on the training dataset.

The CNN models are provided with simulation images to detect the traffic condition, as shown in Figure 9. Here, the images of the vehicles and pedestrians are given to the algorithms separately. The output of the models is the probability for the low, medium, high, and very high categories.

This work uses a feed-forward artificial neural network (ANN) and support vector machine (SVM) model to predict the green light time for a through movement of vehicles at an intersection by using the sensor data as input. It is assumed that the sensor provides the number of vehicles and pedestrians as an input to the algorithms. The output from the models is the green light timings for the ongoing phase. For the ANN model, one hidden layer with five neurons was selected, as shown in Figure 10. The following equations describe the ANN model parameters.
(7)$$u_{k}=\sum_{j=1}^{2}w_{kj}x_{j}$$
(8)$$y_{k}=\varphi \left(u_{k}+b_{k}\right)$$
where $u_{k}$: linear combiner output;$w_{kj}$: weights of neuron *k*;$x_{j}$: inputs to the neural network;$y_{k}$: output of *k*th neuron;φ: activation function;$b_{k}$: bias.

**Figure 10 sensors-23-05028-f010:**
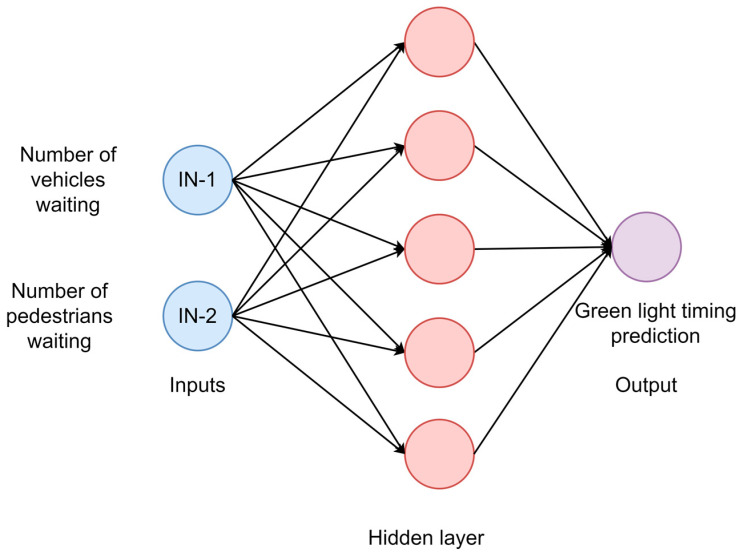
Feed-forward ANN architecture.

The SVM model maps the input data into high dimensional space via mapping the function $f\left(x\right)$. The following is the equation for the SVM model.
(9)$$f\left(x\right)=\langle w,x\rangle +b$$
where $f\left(x\right)$: mapping function;x: input vector;w: weight vector perpendicular to the higher plane;b: bias term.

The predicted data are then stored in a DDE client. Dynamic data exchange (DDE) is a protocol that allows communication between applications. Here, the DDE client is the Excel file where the data are stored. In this work, to initiate the conversation with the DDE server and the ladder logic, a VBA code was used. The client then connects with the control system and sends the input data to the DDE server to adjust the traffic light timings.

#### 3.2.3. Control System

The control system consists of a DDE server that receives the data from the cyber system. The process begins with the Excel sheet requesting data to be entered in the ladder logic. The server receives the request and retrieves the specific instruction from its memory where the data need to be updated. The controller then uses the data in the ladder logic to change the traffic light timing. Whenever the data in the Excel file change, an update is sent to the ladder logic, ensuring that the controller sending the signal to the traffic light is always up to date.

A ladder logic program is utilized to implement the dynamic traffic interval strategy. It is a programming language used in programmable logic controllers (PLCs) to create control systems for industrial processes and machines. The language is based on ladder diagrams, which represent the physical components of the control system, such as traffic lights, and the logic that connects them.

The ladder logic represents the control system as a series of rungs on a ladder. Each rung consists of one or more input conditions that must be met for the output to be activated. The input conditions are connected by logical operators such as AND, OR, and NOT, which allow complex logical operations to be performed. The output of one rung can be connected to the input of another rung, allowing for more complex control systems to be created.

In this work, the ladder logic is separated into main routines and sub-routines. The main routine is the primary program that controls the overall operation of the system. It consists of a series of rungs that are executed sequentially, with each rung controlling a specific sub-routine, as shown in Figure 11. The sub-routines represent the timing operation of the low, medium, high, and very high categories, as shown in Figure 12. Sub-routines can be called from within the main routine, allowing them to be used as building blocks to create more complex control systems.

## 4. Simulation and Modelling

A simulation model was built in the Simulation of Urban Mobility (SUMO) to mimic the real-world intersection and the dynamic traffic interval strategy. It is a widely used open-source microscopic traffic simulation software used for various research purposes [37]. SUMO is primarily used for modeling and analyzing different traffic light control strategies. Researchers use SUMO to simulate different strategies, such as fixed-time or actuated control, and then compare their performance in various metrics, such as average wait time or fuel consumption. This can help understand the relative performance of different strategies and inform the design and operation of traffic light systems.

This research uses a common four-way intersection in the SUMO model to demonstrate the dynamic traffic light logic shown in Figure 13. This intersection has eight phases, where the left turn movements and the through movements on the same road are divided into separate phases. The traffic light phases follow the National Electrical Manufacturers Association (NEMA) convention. Each incoming road is divided into four lanes, and the outgoing road is divided into three lanes. There is a sidewalk for every road that only pedestrians can use. The pedestrians use the crosswalks at the intersection during the through-movement phases.

### 4.1. Parameters

The simulation model defined the traffic flow for each lane to represent real-world traffic scenarios. The highest critical volume used in the simulations was 1000 passenger cars per hour per lane. Additionally, the flows were defined to allow vehicles or pedestrians coming from one direction to travel in all directions. The dimensions of the intersections were default values generated by SUMO. The length of the edges was 150 m. The passenger vehicles were considered to have a length of 5 m, and the minimum gap between vehicles was presumed as 2 m when the vehicles were fully stopped at the intersection. The speed limit was 25 mph for vehicles with a 10% deviation in speed. The pedestrians were considered adults with an average speed of 1.06 m/s with a speed deviation of 10%. Additionally, they would occupy a 0.3 m^2^ area while waiting at the curbside [38]. The detection zone length for the through-movement lanes was 61 m. The pedestrian detection zone was defined before the area where the pedestrian stops, as SUMO has no default option to place the lane area detectors where the pedestrians stop. For turning-movement lanes (left and right), the volume of vehicles was considered as 75% of the through movement lanes. Using the parameters for vehicles and pedestrians, the number of vehicles and pedestrians waiting at the intersection was calculated for low, medium, high, and very high categories, as shown in Table 3. The parameters were kept the same for all traffic flows coming from all directions.

### 4.2. Traffic Distributions

To generate a certain category of traffic flow in the simulation, this work uses the Poisson and binomial distribution for a vehicle flow approaching an intersection. Past line of work [39,40,41,42] shows that the Poisson probability distribution can be used to approximate the behavior of low traffic flow, and the binomial probability distribution can be used to estimate the high traffic flow. Table 4 contains Poisson probabilities for the low and medium traffic flow, whereas the binomial probabilities were used to model the high and very high traffic flow. Similarly, the Poisson probabilities were used to model the pedestrians approaching an intersection. These probabilities would generate four different traffic conditions for vehicles and pedestrians.

### 4.3. Dynamic Traffic Light Algorithm

This work uses the Simulation of Urban Mobility (SUMO) Traffic Control Interface (TraCI) module that controls the SUMO model using dynamic traffic logic. It allows the retrieval of item values and the live modification of their behavior by granting access to a running road traffic simulation [37]. Algorithm 1 shows the pseudo-code for adjusting the green traffic light timing based on the detected number of pedestrians and traffic detected for the through traffic phase.
**Algorithm 1** Dynamic Traffic Light adjustment1Input:*n* ← number of through phases2
*i* ← current through phase3
*vi* ← detected vehicles4
p*i* ← detected pedestrians5
t*_i_* ← traffic light timing6Output:W_1_ ← average vehicle delay7
W_2_ ← average pedestrian delay8Procedure:AdjustTrafficLightTiming9
p*i* = GetHaltedPedestrians (Tr → Condition)10
v*i* = GetHaltedVehicles (Tr → Condition)11
**for** *i*= (1 *to* *n*) do12
**if** Tr = low **then**13
   t*_i_* = low14
  **elseif** Tr = medium **then**15
   t*_i_* = medium16
  **elseif** Tr = high **then**17
   t*_i_* = high18
  **else** Tr = very high **then**19
t*_i_* = very high20
**end if**21
**end for**22
**Output:** W_1_, W_2_

Figure 14 shows the architecture of the dynamic traffic light interval control strategy. At the beginning of each phase, the sensors input the dynamic traffic logic, and the green light timing will be altered according to the dynamic logic. This research utilizes a combination of pedestrian and vehicle green time in the simulation to reduce the waiting time for automobiles and pedestrians. The TraCI module interacts with the simulation online, whereas the communication between the TraCI and the cyber system happens in fixed iterations.

### 4.4. Design of Experiments

To check the performance of the dynamic control logic, it was compared with the pre-timed and semi-dynamic traffic systems. The simulation parameters were the same for all the traffic light control strategies except the traffic light timings. In the pre-timed traffic light system, the green light timings were fixed irrespective of the traffic condition. The green light timings for each phase were set to allow the highest volume to pass through the intersection without green light extension. So, the pre-timed traffic light system used the timing of the very heavy traffic condition category. For the semi-dynamic traffic light system, the traffic light timings were divided into two categories: rush hours and non-rush hours. So, here, the timings of medium and very heavy traffic conditions categories were used for the green light during non-rush hours and rush hours, respectively. The traffic demand was built in a way to mimic real-world traffic conditions for 24 h. Table 5 gives a detailed breakdown of the vehicle and pedestrian volumes at different times of the day.

The distribution of the traffic flow was based on the peak hours for urban areas from 6 am to 10 am and 4 pm to 8 pm as stated by FHWA [43], where the vehicle and pedestrian categories used around this time are medium, high, and very high. Additionally, to establish demand at an intersection, the travel time for Houston was considered over a period of day for the year 2022 [44]. The demand was built to test the control strategy for different combinations of vehicle and pedestrian volume. This demonstrates the variability in the volume of vehicles and pedestrians over the course of a day. In the simulations, the overall waiting time was checked and compared for pre-timed, semi-dynamic, and dynamic traffic systems.

## 5. Data Analysis and Findings

Simulations were conducted using the parameters given in Table 6 to check the effect of pedestrian volume on vehicle waiting time and vehicle volume on pedestrian waiting time. In total, 160 simulations were conducted, where ten samples were taken for each combination of pedestrian and vehicle volume. The pedestrian volume was changed, and the average waiting of the vehicles and pedestrians was noted over the one-hour simulation period with an average of 30 traffic cycle lengths, keeping the vehicle’s volume constant. All the parameters were kept same in one combination set of vehicle and pedestrian volume, while changing the seed value for each simulation. This would generate slightly different spawning patterns of vehicles and pedestrians in the simulation.

A one-way ANOVA (analysis of variance) test was performed to compare the difference of mean among the various combinations of traffic category groups as listed. It is a statistical method used to analyze the differences between the means of three or more independent groups based on a single categorical independent variable. A one-way ANOVA aims to determine whether there is a significant difference among the means of the groups, suggesting that the factor has a significant effect on the continuous dependent variable. Table 7 lists the vehicle waiting time for pedestrian volume change, and Table 8 lists the pedestrian waiting time for vehicle volume change.

The results show that the *p*-value for each group combination is less than the 0.05 significance level. Thus, the null hypothesis is rejected, and there is a difference in the mean of the groups. Similarly, for the pedestrian waiting time, the *p*-value for each group combination is less than the significance level, indicating that the change in user volumes affects the other users at the intersections. Additionally, an examination of the average waiting times reveals that the traffic waiting time increases as the volumes change from low to high. The results are given in Table 9 and Table 10.

The machine learning performance was also tested. The ANN and SVM models were given 1200 data samples to predict the green light timing for detected traffic conditions. Out of all the samples in the dataset, 70% of the data were used for training, and 30% were used in testing the neural networks. Table 11 shows the mean square error (MSE), R^2^ value, average training, and testing accuracy of both models. The results indicate that both models performed well, with the SVM model performing slightly better than the ANN model.

SVM’s performance was better in this case because the relationship between the input and output data was fairly linear so that the model could find an optimal decision boundary, called the maximum–margin hyperplane, which tends to generalize well to unseen data. Additionally, the models did not need to predict any complex patterns in the traffic light timing. Moreover, SVM models tend to be more interpretable than ANNs, as they involve finding a hyperplane that separates the data. This can be important when the goal is not only to make predictions but also to understand the underlying patterns in the data.

On the other hand, the ANN model accuracy was very close to the SVM model accuracy. However, ANN models are more vulnerable to overfitting, which may reduce the accuracy of the ANN model in predicting the output.

Ten different pre-trained CNNs were used to predict the traffic condition from images. The pre-trained models are typically trained on a large dataset and have learned valuable features that can be reused in a new task. The training and testing accuracies of all the networks are listed in Table 12. The results show that almost all the models performed well in detecting the vehicles. Darknet53 achieved the highest accuracy of 97.5% validation accuracy compared to others. Darknet53 is a CNN model that has a total of 53 layers. This model performs well in detecting the objects from the given image dataset. Similarly, Darknet19, Alextnet, and Inceptionv3 are known for their high accuracy in object recognition.

The performance of the different CNN models depends on their architecture. Different architectures have unique strengths and weaknesses. Resnet18 is designed to handle deeper networks by introducing skip connections, while Mobilenetv2 uses depth-wise separable convolutions for efficient computation. More complex models such as Resnet101 can capture more intricate patterns but may require more computation and be prone to overfitting. Simpler models such as SqueezeNet and Mobilenetv2 are designed for efficiency, trading off some accuracy for reduced computational requirements. Thus, their validation accuracies are 47.5% and 80%, respectively, which are the least compared to other models. 

However, the validation accuracy of all models plunged for pedestrian detection cases. This happened because of the incorrect data from the simulation. Some images of the pedestrian lanes taken during the simulation represented more pedestrians than the detected category. Pedestrians can move in two directions on a single lane, whereas the vehicle forms a proper queue and moves in a single direction only on a single lane. So, while capturing the images of the pedestrians looking towards the signal, the camera also captured the pedestrians moving in the opposite direction of the traffic signal.

The method proposed in this work was compared with the time-gap-based and delay-based traffic light actuation methods [45,46]. These two methods are established methods and have been proven to reduce traffic delays at intersections in real settings and in simulations. Table 13 compares the average waiting time of vehicles for 12 h simulations for low, medium, high, and very high volumes of vehicles. In these simulations, pedestrian flows were not considered. From the table, some instructive points can be noted. The time-gap-based and delay-based traffic light control strategy performed better for the low and medium volume of vehicles, whereas the waiting time was similar for a high traffic volume. The dynamic traffic interval technique presented in this work outperformed the others under the condition of very high traffic volume. This is because, in the dynamic traffic intervals technique, the times are fixed for a detected volume of traffic for a given phase in a cycle. Thus, even if there are more cars coming at an intersection after a certain gap or time delay, the signal will remain green for the fixed period.

The dynamic traffic light interval technique was also tested alongside fixed time and semi-dynamic traffic light plans to evaluate the performance in terms of the delay. In these simulations, the pedestrian and vehicle flows were considered. Twenty-five simulations were conducted to test out the proposed technique thoroughly. The experimental design was structured to adjust the categories according to the time of day. The fixed-time traffic light plan used very-high-category-traffic timing, and it remained fixed irrespective of the change in input probabilities. Similarly, the semi-dynamic traffic plan used the timings of medium- and very-high-category traffic. The time in semi-dynamic traffic lights would change based on the rush and non-rush hours of the day.

Based on the simulation results, there was a 20.11% to 26.67% reduction in the overall vehicle waiting time for an intersection compared to the fixed-time traffic light system and an 11.99% to 18.09% reduction compared to the semi-dynamic traffic light system. Additionally, a 20.38% to 23.16% drop was observed in the overall pedestrian waiting time at an intersection compared to the fixed-time traffic light system and an 8.55% to 10.95% reduction compared to the semi-dynamic traffic light system. Table 14 shows the performance comparison for the three different traffic light control strategies.

## 6. Conclusions and Future Work

This work proposed a system that simultaneously reduced the waiting time of vehicles and pedestrians by categorizing the vehicle and pedestrian volume into categories. The model achieved this result by adjusting the green light timings of the ongoing phase. Reducing the waiting time of vehicles at the intersection means indirectly lowering vehicle emissions, reducing fuel consumption, and effectively utilizing the traffic signal cycle length. Moreover, lowering the delays for pedestrians decreases the chances of the dangerous behavior of people trying to cross the roads.

This work also demonstrated a framework to implement the dynamic traffic interval strategy. The cyber system, using CNN, proved its usefulness in detecting traffic conditions at the intersection. ANN/SVM helped to set the timing for the combination of the pedestrian and vehicle traffic condition. Moreover, ladder logic programming controlled the traffic light timings based on the traffic condition detected. In this model, all vehicles were assumed as passenger cars having an equal length, and each car approaching the intersection maintained the same distance from the other. Additionally, the simulation considered the same critical volume for all through movements of vehicles. As this technique is dependent on the service requested by the users, it will not be applicable at all intersections. This is particularly true in areas where the demand is very low, where few pedestrians are requesting a service, or when traffic variation is minimal over the course of the day.

In the future, more parameters can be provided as inputs to machine learning models to predict traffic conditions accurately. Moreover, green light timing was calculated based on the number of vehicles and pedestrians. In the future, more parameters will be considered, such as pedestrians and vehicle speed, and other vehicles, such as trucks, buses, and motorbikes, to adjust the green light time. One crucial assumption in this research was that the cameras provide all the data to the cyber system. However, this may not be possible in all scenarios; thus, data from different sensors such as ultrasonic and infrared will be considered in the future. Additionally, this work focused on the working of signal intersections. So, future work will include a control strategy considering multiple coordinating intersections.

## Figures and Tables

**Figure 1 sensors-23-05028-f001:**
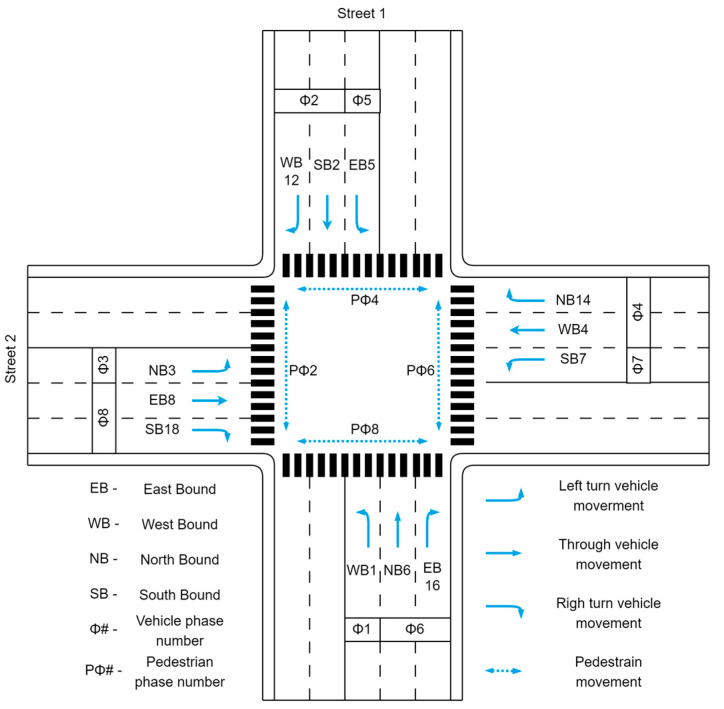
Traffic movement at an intersection.

**Figure 2 sensors-23-05028-f002:**
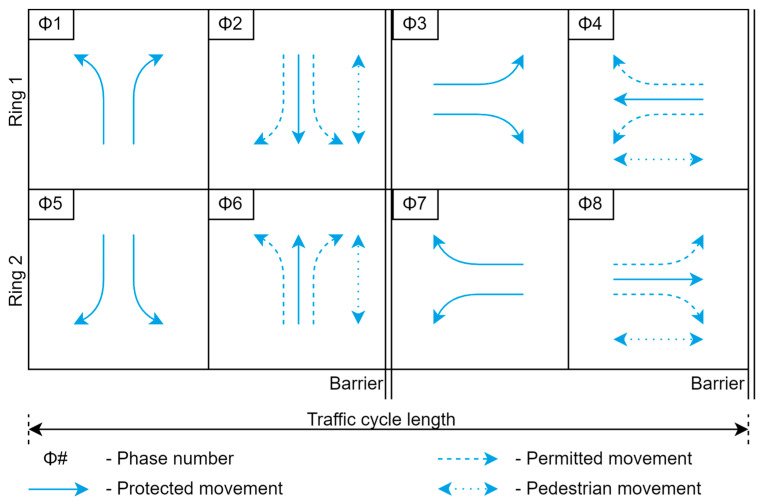
Ring and barrier diagram.

**Figure 3 sensors-23-05028-f003:**
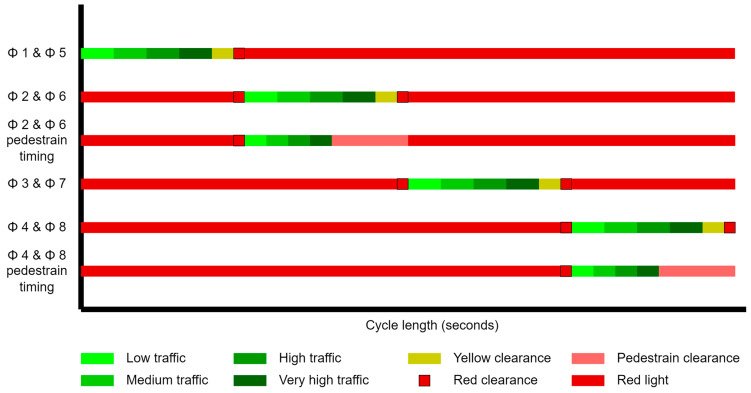
Traffic light timing distribution.

**Figure 5 sensors-23-05028-f005:**
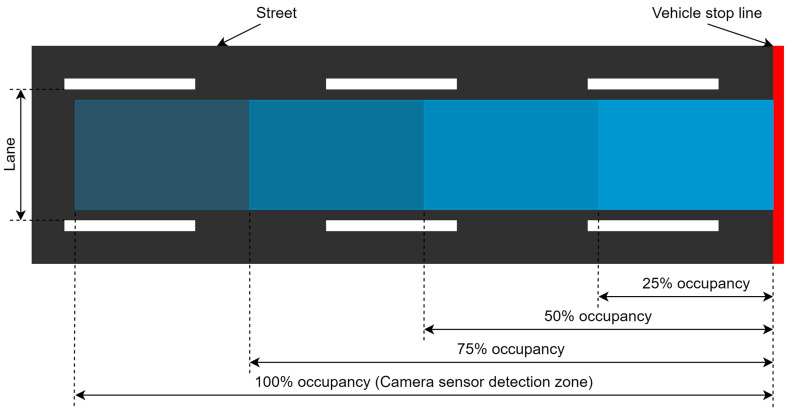
Vehicle occupancy zone.

**Figure 6 sensors-23-05028-f006:**
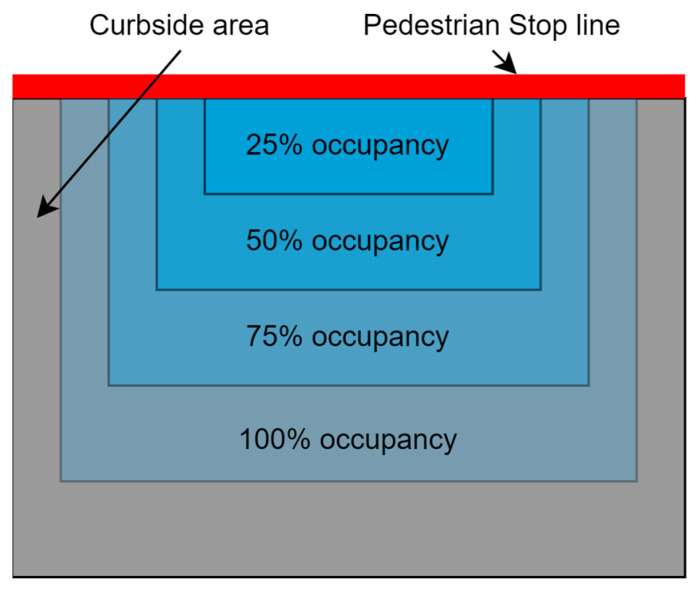
Pedestrian occupancy zone.

**Figure 7 sensors-23-05028-f007:**
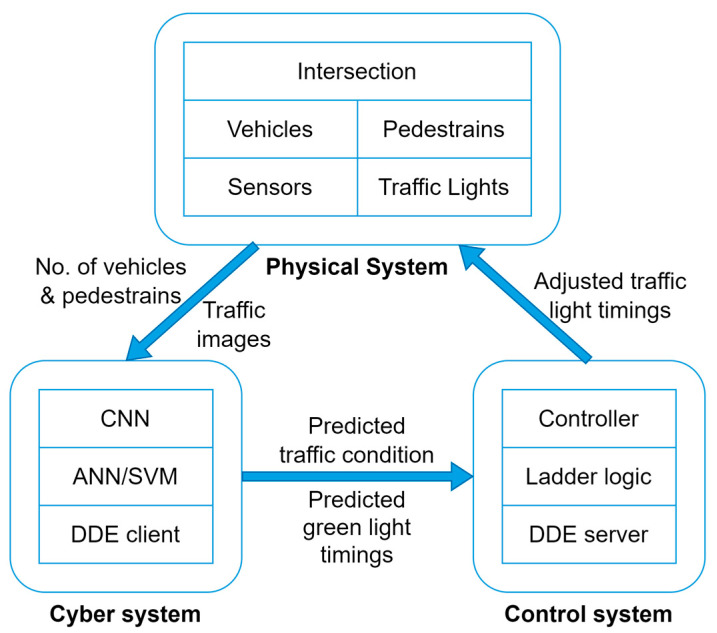
Overview of Cyber-Physical System.

**Figure 8 sensors-23-05028-f008:**
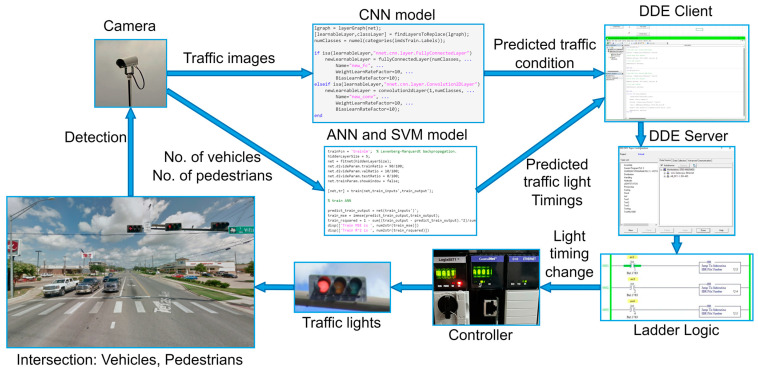
Operation of smart traffic light control system.

**Figure 9 sensors-23-05028-f009:**
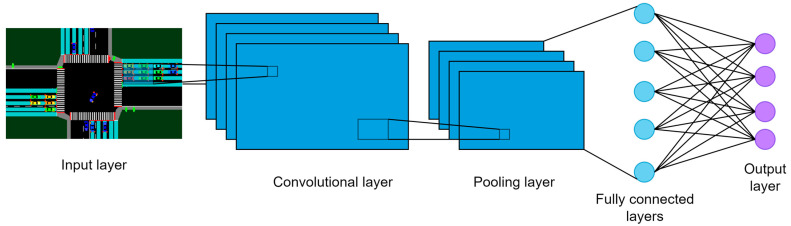
CNN architecture.

**Figure 11 sensors-23-05028-f011:**
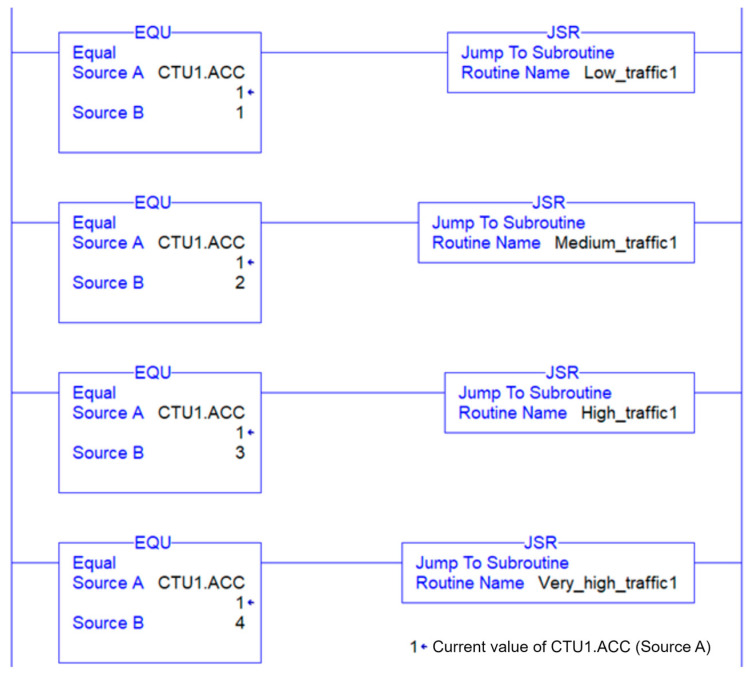
Ladder logic program—Main routine.

**Figure 12 sensors-23-05028-f012:**
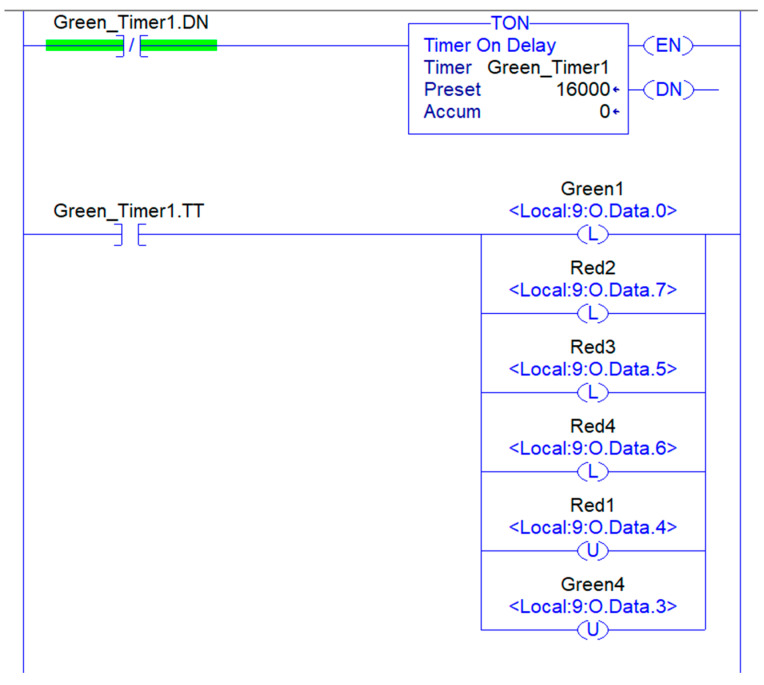
Ladder logic program—Sub routine.

**Figure 13 sensors-23-05028-f013:**
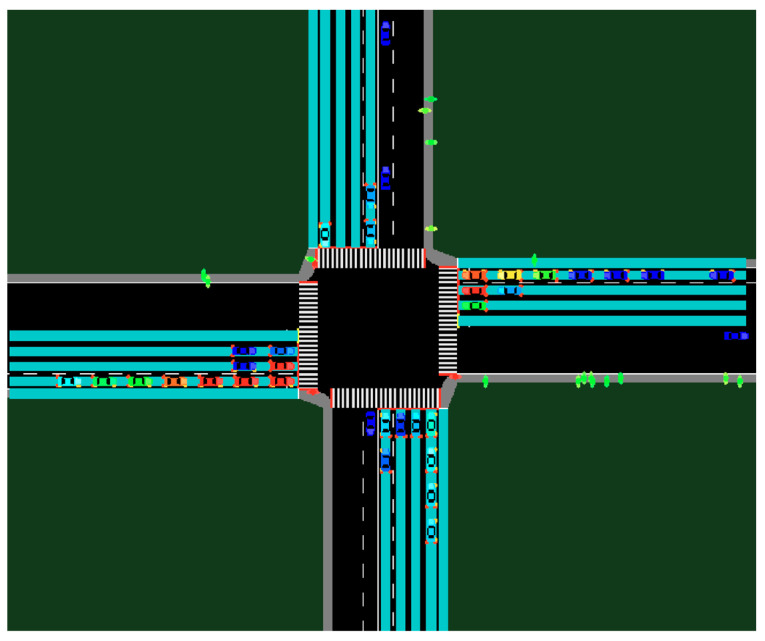
A 4-way, 8-phase intersection.

**Figure 14 sensors-23-05028-f014:**
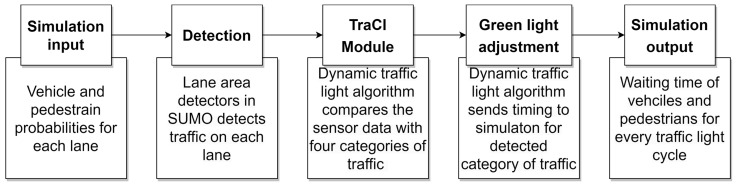
Dynamic traffic light algorithm.

**Table 1 sensors-23-05028-t001:** Comparison of various intelligent traffic light systems.

Previous Work	Sensor(s) Type Mentioned	Controller Type Mentioned	Simulation Platform Mentioned	Machine Learning Used	Results Presented	Categorization of Traffic
Ghazal et al. [5]	IR sensors	PIC 16F877A microcontroller	No	Yes	No	Yes: high density and low density
Diaz et al. [6]	PIR sensors	Raspberry Pi	No	No	No	No
Silva et al. [7]	Cameras, radars	No	No	No	Yes	No
Alharbi et al. [8]	WSN, RFID tags	No	MATLAB	No	Yes	No
Younis and Moayeri [11]	Piezoelectric (proposed)	Raspberry Pi 3 (proposed)	C++	No	Yes	Yes: light traffic and heavy traffic
Tchuitcheu et al. [12]	WSN	No	SUMO	No	Yes	No
Gandhi et al. [13]	Cameras	No	Python, Pygames	Yes	Yes	No
Pratama et al. [14]	IR sensors	PIC 16F876A microcontroller	N/A	No	Yes	No
Chavan et al. [15]	IR sensors	AT 89C51 microcontroller	Microcontroller assembly language	No	Yes	No
Hirankitti et al. [16]	No	No	NetLogo simulator	No	Yes	No
Almawgani [17]	Camera	Arduino	No	No	Yes	No
Li et al. [18]	UAV/ cameras	No	SUMO	Yes	Yes	No
Wiering et al. [19]	No	No	Green Light District simulator	Yes	Yes	No
Wei et al. [20]	Camera	No	C++, MATLAB	Yes	Yes	No
Liu et al. [21]	Cameras	No	SUMO	Yes	Yes	No
Linag et al. [22]	Vehicular/ Sensor networks	No	SUMO	Yes	Yes	No
Genders and Razavi [25]	No	No	SUMO	Yes	Yes	No
Oliveira et al. [26]	No	No	SUMO	Yes	Yes	Yes: small, medium, and large vehicles
Abbas et al. [27]	No	No	SIRDA intersection simulator, MATLAB	No	Yes	No
McKenney and White [28]	No	No	SUMO	No	Yes	No
Zhou et al. [29]	WSN	No	No	No	Yes	No
Piris et al. [30]	WSN	No	SUMO	No	Yes	No
Navarro-Espinoza et al. [31]	No	No	No	Yes	Yes	No
Muntean [32]	WSN	No	No	Yes	Yes	No
Artega et al. [33]	On-road sensors	No	SUMO	No	Yes	Yes: very low, low, medium, high, and very high
Nimac et al. [34]	Radar	No	No	No	No	No

**Table 2 sensors-23-05028-t002:** Traffic categories and corresponding occupancy.

Traffic Category	Occupancy
Low traffic	0–25%
Medium traffic	26–50%
High traffic	51–75%
Very high traffic	76–100%

**Table 3 sensors-23-05028-t003:** Parameters for traffic categories.

Parameters		Traffic Conditions			
		Low	Medium	High	Very High
Critical volume (vehicles/h/lane)		416	718	920	1071
Detection zone	Vehicle (length) (m)	16	32	48	61
	Pedestrian (area) (m^2^)	1.86	3.72	5.58	7.44
No. of vehicles	Left vehicles	0–1	2–3	4–5	6–7
	Through vehicle	0–2	3–4	5–6	7–8
No. of pedestrians		0–2	3–4	5–6	7–8
Green Light timing (s)	Pedestrian	4	5	6	7
	Left movement	9	12	15	19
	Through movement	16–19	16–19	20	26

**Table 4 sensors-23-05028-t004:** Probabilities for four traffic conditions.

Traffic Conditions	Lane Probabilities (Vehicles/s)		Pedestrian Probability (Pedestrians/s)
	Through Vehicle	Left Vehicle	
Low	0.033	0.024	0.012
Medium	0.057	0.042	0.024
High	0.065	0.049	0.035
Very High	0.079	0.059	0.047

**Table 5 sensors-23-05028-t005:** Vehicle and Pedestrian volume for 24 h simulation.

Time of the Day	Traffic Conditions	
	Vehicles	Pedestrians
12:00 a.m.–5:00 a.m.	Low	Low
5:00 a.m.–6:00 a.m.	Low	Medium
6:00 a.m.–7:30 a.m.	Medium	Medium
7:30 a.m.–9:00 a.m.	Very high	High
9:00 a.m.–10:30 a.m.	Very high	Very high
10:30 a.m.–12:00 p.m.	High	Medium
12:00 p.m.–1:30 p.m.	Medium	High
1:30 p.m.–3:00 p.m.	Low	Very high
3:00 p.m.–4:30 p.m.	Low	High
4:30 p.m.–6:00 p.m.	Medium	Very high
6:00 p.m.–6:45 p.m.	High	High
6:45 p.m.–7:30 p.m.	High	Very high
7:30 p.m.–8:15 p.m.	Very high	Medium
8:15 p.m.–9:00 p.m.	Very high	Low
9:00 p.m.–10:30 p.m.	High	Low
10:30 p.m.–12:00 a.m.	Medium	Low

**Table 6 sensors-23-05028-t006:** Parameter settings for hypothesis test simulations.

Volume Category		Traffic Light Timing		
Vehicle	Pedestrian	Through Vehicles	Pedestrians	Left Turn Vehicles
Low	Low	12	4	9
	Medium	12	5	9
	High	12	6	9
	Very high	12	7	9
Medium	Low	12	4	12
	Medium	12	5	12
	High	12	6	12
	Very high	12	7	12
High	Low	16	4	15
	Medium	16	5	15
	High	16	6	15
	Very high	16	7	15
Very high	Low	22	4	19
	Medium	22	5	19
	High	22	6	19
	Very high	22	7	19

**Table 7 sensors-23-05028-t007:** Effect of pedestrian volume change on vehicle waiting time.

Vehicle Volume	Pedestrian Volume	Vehicle Waiting Time (s)										Average	Std. Dev.
Low	Low	12.3	11.4	12	12.4	12.3	12.2	12.1	12.3	12.2	11.7	12.084	0.31711
	Medium	13.1	12.4	12.8	12.6	12.7	12	12.1	11.7	12.1	12.7	12.422	0.45161
	High	13.1	13.8	13.6	12.9	13	12.5	12.8	12.9	13	13	13.065	0.37041
	Very high	13.3	13.7	13	13.5	13.2	13.4	13	13.8	13.6	13	13.334	0.30537
Medium	Low	16.4	16.3	15.9	15.7	16.7	15.9	15.2	15.9	15.7	15.1	15.863	0.50953
	Medium	16.4	16.2	15.7	17	16.1	15.6	16.1	17	17	16.6	16.368	0.52434
	High	16.1	17.4	17	16.5	17.5	17	16.4	16	16.3	17.7	16.787	0.61056
	Very high	17.5	17.2	16.5	16.4	17.7	16.9	17.2	16.5	16.5	18	17.037	0.55331
High	Low	19.4	18.5	19	20.1	19.4	19.3	18.6	18.8	19.2	18.6	19.077	0.48585
	Medium	19.9	21.2	20.6	19.2	20	20.7	20.4	20.1	20	20.1	20.207	0.55542
	High	20.8	21.1	22	20.1	19.9	19.7	19.9	20.9	20.6	21.5	20.63	0.75673
	Very high	20.9	19.6	19.9	21.3	21.9	21.7	22.3	21.8	20.1	21.1	21.062	0.91183
Very high	Low	28.6	29.8	26.8	28.2	25.6	27.5	26	32.5	28.1	27.9	28.095	1.9663
	Medium	32.6	27.5	30.8	28.2	26.3	32	27.1	30.1	32.4	29.8	29.668	2.31398
	High	32.1	28.1	33.3	33.2	30.8	30.3	32.5	31.5	29.2	31.3	31.227	1.68779
	Very high	32.1	31.6	30.7	31.1	30.2	32.9	33	33.3	28.8	31.6	31.516	1.40426

**Table 8 sensors-23-05028-t008:** Effect of vehicle volume change on pedestrian waiting time.

Pedestrian Volume	Vehicle Volume	Pedestrian Waiting Time (s)										Average	Std. Dev.
Low	Low	29.1	26.3	26.2	27.7	29.4	27.9	28.3	27.7	27.9	27.9	27.825	0.97328
	Medium	31	28.5	28.7	30.6	29.8	29.1	28.5	27.5	29.3	30.5	29.343	1.04039
	High	36	33.4	34.3	36.7	35.4	34.4	34.4	33.9	33.8	35.1	34.737	0.97946
	Very high	42.2	44.3	41.8	41.9	40.1	41	41	39.7	42.3	41.3	41.547	1.21186
Medium	Low	29.6	30.1	29.9	30	30.6	30.4	30	27.5	29.3	28.4	29.566	0.92792
	Medium	31.1	30.7	30.7	32.2	31.7	31.2	32.2	32	30.7	31.4	31.381	0.58911
	High	35.9	35.1	36	35.2	36.8	37.7	37.1	36.2	35.5	36	36.154	0.80135
	Very high	42.8	43.3	42.4	41.8	42.9	43.8	40.7	42.3	42.8	44.6	42.742	1.0153
High	Low	32.2	31.6	28.3	29.9	30	31.5	30.9	30.4	30	30.6	30.539	1.0476
	Medium	32.7	33.5	32.5	32.7	32.2	32.3	32.8	32.3	32.1	32.7	32.565	0.37824
	High	37	37.8	37.4	38.4	36.8	38.2	36	35.9	37.5	38.6	37.349	0.90547
	Very high	43	42.6	44.8	44.8	44	44.7	42.3	44	45.8	44.2	44.019	1.0458
Very high	Low	31.7	30.6	31.3	31.4	31.4	31	31.2	30.4	31.7	32.2	31.28	0.49786
	Medium	33.6	33.8	35.3	33.5	33.8	33.4	33.5	33.8	33.6	43.3	34.748	2.88001
	High	39.2	37.4	38.5	40	38.4	41	38.6	39.2	39.5	39.5	39.137	0.94161
	Very high	47	45.4	44.8	43.6	45.5	46.3	46.2	45.4	45.3	46.6	45.619	0.92143

**Table 9 sensors-23-05028-t009:** Results of ANOVA test for vehicle waiting time.

One-Way ANOVA Test	F	*p*-Value	F Critical	Hypothesis Result
Low vehicle vs. variable pedestrian volume	24.65366825	7.58 × 10^−9^	2.866265551	Reject null hypothesis
Medium vehicle vs. variable pedestrian volume	8.715096705	0.000177	2.866265551	Reject null hypothesis
High vehicle vs. variable pedestrian volume	14.92596812	1.79 × 10^−6^	2.866265551	Reject null hypothesis
Very high vehicle vs. variable pedestrian volume	7.101865068	0.000721	2.866265551	Reject null hypothesis

**Table 10 sensors-23-05028-t010:** Results of ANOVA test for pedestrian waiting time.

One-Way ANOVA Test	F	*p*-Value	F Critical	Hypothesis Result
Low pedestrian vs. variable vehicle volume	311.4556	8.51 × 10^−6^	2.866266	Reject null hypothesis
Medium pedestrian vs. variable vehicle volume	432.7093	2.77 × 10^−28^	2.866266	Reject null hypothesis
High pedestrian vs. variable vehicle volume	409.7162	7.21 × 10^−28^	2.866266	Reject null hypothesis
Very high pedestrian vs. variable vehicle volume	133.925	1.39 × 10^−19^	2.866266	Reject null hypothesis

**Table 11 sensors-23-05028-t011:** Performance of ANN and SVM models.

Neural Networks	Training Accuracy	Validation Accuracy	MSE	R^2^
ANN	97.897%	97.678%	0.44834	0.93813
SVM	98.557%	98.191%	0.47739	0.93412

**Table 12 sensors-23-05028-t012:** Performance of CNN models.

Pre-Trained Networks	Vehicle Detection		Pedestrian Detection	
	Training Accuracy	Validation Accuracy	Training Accuracy	Validation Accuracy
squeezenet	46.875%	47.5%	36.7188%	42.5%
googlenet	69.5313%	82.5%	54.6875%	57.5%
resnet18	92.1875%	90%	73.4375%	55%
mobilenetv2	88.2813%	80%	78.1250%	62.5%
resnet50	92.1875%	82.5%	85.9375%	57. 5%
resnet101	100%	85%	100%	60%
inceptionv3	98.4375%	90%	95.3125%	53.5%
alexnet	87.5%	92.5%	75%	67.5%
darknet19	100%	90%	100%	60%
darknet53	100%	97.5%	100%	55%

**Table 13 sensors-23-05028-t013:** Waiting time of vehicles for three different traffic light control strategies.

Traffic Volume	Dynamic Traffic Intervals (s)	Gap-Based (s)	Delay-Based (s)
Low	12.13	9.72	9.09
Medium	13.81	13.22	12.27
High	15.06	15.77	15.06
Very high	20.19	28.46	35.48

**Table 14 sensors-23-05028-t014:** Performance comparison of different traffic light control strategies.

Technique	Vehicle Waiting Time (s)				Pedestrian Waiting Time (s)			
	Average	Std. Dev.	Minimum	Maximum	Average	Std. Dev	Minimum	Maximum
Fixed time	24.58	0.5616	23.93	26.25	42.36	0.1533	42.22	42.53
Semi-dynamic	22.03	0.5171	21.25	23.5	36.16	0.1654	36.63	37
Dynamic	18.77	1.4653	18.28	19.25	33.16	0.2576	32.76	33.33

## Data Availability

The data presented in this study are openly available in FigShare at https://doi.org/10.6084/m9.figshare.22689676, reference number 22689676 (accessed on 12 May 2023).

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
