# Peer review of "Cyber-Physical System for Smart Traffic Light Control"

_sensors, 2023, doi:10.3390/s23115028_

Round 1

Reviewer 1 Report

The paper presents a scheme for deploying a traffic signal system intended to dynamically adjust the phases for pedestrians and vehicle flows, minimizing the waiting times for both actors. The paper presents a literature review regarding sensor-based and machine-learning controllers and insights into theoretical and empirical results. However, I have some concerns about the manuscript:

1. It is not clear how the saturation flow is estimated to accomplish equation (4).

2. Regarding the implementation of the Webster Method for signalization, it is not clear whether the four estimated cycle lengths are applied to adaptively reprogram the traffic signal or to establish fixed extensions to perform an actuated scheme.

3. The section where the cyber system is described lacks detail. The authors omitted details about how the images are represented to be useful as inputs for the CNN and how the data is refined through the hidden layers.

4. The presentation of the simulation model also has several details that lack clarity. Firstly, it is not specified how the simulation was calibrated. In this sense, the authors are encouraged to provide details about how the virtual scenario trustily represents the real-world traffic flow, according to the case study, statistical principles, and time of days. As an aside, the manuscript mentions that the TraCi API is used, but it is not stated whether the API interacts in an online manner with the cyber system or with fixed estimations. Finally, regarding the results collection, the manuscript states that the waiting time was noted over the one-hour simulation period of 30 traffic light cycles, however, it is confusing whether each one-hour simulation was carried out regardless of other configurations (others one-hour simulations) or whether the different intervals of one hour are dependent on the previous ones. This is relevant due to the way the road network is filled and how the whole system transit to a steady state. Overall, the authors are encouraged to provide richer details that could help the reader to understand why the simulation experiments have confidence with respect to the behavior of the real-world traffic network. 

5. A better performance comparison is needed to trustily support the results. The manuscript focused the analysis by only considering a pre-timed scheme and a semi-dynamic one that is not completely explained. As stated by the authors, there is a lack of traffic signal controllers simultaneously regulating vehicles and pedestrian flows. Nevertheless, since traffic signal control for isolated and coordinated intersections is a widely studied topic, a novel proposal needs to demonstrate to be effective and efficient against the previously published ones. In this way, the authors must consider the fair comparison against some already accepted approaches, to demonstrate, in the first stage, that the proposal is effective to regulate vehicle streams to better support the proposal by integrating a pedestrian flow regulation in the second stage. In this way, the authors may consider the comparison against the time-delay and time-gap approaches include in SUMO [1, 2]. Those solutions have been demonstrated to be two of the most competitive solutions in simulation environments and also in real-world implementations, including isolated and coordinated intersections.

6. The simulation traffic levels seem low, especially considering that the saturation flow in arterial roads commonly surpasses 1750 veh/h. This last is critical to judge the effectiveness and generality of the proposal since the approach is based on the green time extension strategy, which tends to produce large cycle lengths when traffic becomes crowded [3, 4].

7. A more affordable explanation about the use of the linguistic terms “low”, “medium”, and “very high” is needed. Commonly this kind of parametric fashion is used in signal controllers based on fuzzy logic, where such terms are closely related to specific discretized scalar intervals [5, 6]. In the case of this paper, it is hard to understand how these terms are mapped.

[1] Oertel, R.; Wagner, P. Delay-time actuated traffic signal control for an isolated intersection. In Proceedings of the Transportation Research Board 2011 (90th Annual Meeting), Washington, DC, USA, 23–27 January 2011. Number EPFL-CONF-181089.

[2] Erdmann, J.; Oertel, R.; Wagner, P. VITAL: A Simulation-Based Assessment of New Traffic Light Controls. In Proceedings of the 2015 IEEE 18th International Conference on Intelligent Transportation Systems, Gran Canaria, Spain, 15–18 September 2015; pp. 25–29.

[3] Araghi, S.; Khosravi, A.; Creighton, D. A review on computational intelligence methods for controlling traffic signal timing. Expert Syst. Appl. 2015, 42, 1538–1550

[4] Rasheed, F.; Yau, K.L.A.; Noor, R.M.; Wu, C.; Low, Y.C. Deep Reinforcement Learning for Traffic Signal Control: A Review. IEEE Access 2020, 8, 208016–208044

[5] Ali, M.E.M.; Durdu, A.; Celtek, S.A.; Yilmaz, A. An Adaptive Method for Traffic Signal Control Based on Fuzzy Logic with Webster and Modified Webster Formula Using SUMO Traffic Simulator. IEEE Access 2021, 9, 102985–102997.

[6] Madrigal Arteaga, V.M.; Pérez Cruz, J.R.; Hurtado-Beltrán, A.; Trumpold, J. Efficient Intersection Management Based on an Adaptive Fuzzy-Logic Traffic Signal. Appl. Sci. 2022, 12, 6024. 

Author Response

Dear Reviewer,

We greatly appreciate the time you invested in reviewing our manuscript and identifying its flaws. We have carefully considered your comments and have subsequently made revisions to address them.

For a detailed account of our responses, please see the attachment.

Once again, we extend our gratitude for your valuable input.

Sincerely,

Siddhesh Deshpande

Reviewer 2 Report

In this work, the authors proposed a system that simultaneously reduced the waiting time of vehicles and pedestrians by categorizing the vehicle and pedestrian volume into categories. The topic is interesting. There are some problems which should be addressed.

1. I understand that machine learning (ML) has been a promising tool in the smart traffic problems. I am concerned with the reliability of ML because we  can not guarantee it works with100% accuracy. However, any traffic accident is not acceptable in the real world. Could you give some deep discussion about this?

2. Please check the grammar errors.

Generally, the English are OK. Please further polish and corrcect all grammar errors.

Author Response

Dear Reviewer,

We greatly appreciate the time you invested in reviewing our manuscript and sharing your concern on this topic.

Please see the attachment for our responses.

Once again, we extend our gratitude for your valuable input.

Sincerely,

Siddhesh Deshpande

Reviewer 3 Report

This is a simple but interesting work presenting the new strategy of traffic light control using heuristic tools and simulating model. The paper is almost ready for publishing, however, I ask the Authors to consider some elements as below.

1. The literature should be a little fresh and should contain sources from MDPI Journals, especially “Sensors”, “Remote sensing” or “Energies”.

2. The strategy assumes vehicles or pedestrians just waiting before the intersection (in four areas). What with the vehicles or pedestrians approaching an intersection (who wish to cross after a small period)?

3. In what way the destination of pedestrian movement (direction) is detected? Observing the considered corner of the intersection in Figure 13 it is not clear if a specific pedestrian will go to one of two crossings (for example to the north or east) or will veer (not using the crossing).

4. What if the intersection has a simpler structure than in Figure 13? Especially if have a smaller number of lines on the selected approach (with no separation of left or right turns).

5. It is an interesting concept with the use of diagonal pedestrian crossings and specific phases or control strategies. For an example see doi: 10.3390/en15186579 – case 2. Is it possible to use the presented strategy for such a solution? Especially, considering the problems described in points 2-4.

6. The fourth row in the second part of Table 1 (page 5) has no sense. The last column in this table is not necessary because contains the same information for all cases (rows). The lack of consideration of pedestrian data in all of the previous work can be commented on in the text (except if the list of previous works will be enlarged according to my remark 1 or 5).

Minor editing of English language required.

Author Response

Esteemed Reviewer,

We extend our sincerest gratitude for dedicating your valuable time to thoroughly evaluate our manuscript and highlighting its areas of improvement. We have taken the necessary steps to address the mentioned concerns.

For a comprehensive overview of our responses, please see the attachment.

We would like to express our heartfelt appreciation once more for your invaluable contribution.

Warm regards,
Siddhesh Deshpande

Round 2

Reviewer 1 Report

I've revised both the response letter and the modified paper and I consider that all my comments have been attended to, so all my doubts have been clarified. I think the manuscript can be considered for publication.